# Mainstreaming Climate-Smart Agriculture in Small-Scale Farming Systems: A Holistic Nonparametric Applicability Assessment in South Africa

**Victor O. Abegunde [1], Melusi Sibanda [1,*] and Ajuruchukwu Obi [2]**

[1] Department of Agriculture, Faculty of Science & Agriculture, University of Zululand, KwaDlangezwa 3886, South Africa; overcomers001.av@gmail.com

[2] Department of Agricultural Economics & Extension, University of Fort Hare, Alice 5700, South Africa; aobi@ufh.ac.za

* Correspondence: SibandaM@unizulu.ac.za; Tel.: +27-(0)-35-902-6068

**Abstract:** Current research focuses disproportionately on the characteristics of farmers to understand the factors that influence the introduction of climate-smart agriculture (CSA). As a result, there has been a failure to take a holistic view of the range of drivers and barriers to CSA implementation. Many aspects of technologies or practices that may encourage or inhibit the implementation of CSA and define its applicability are, therefore, not systematically considered in the design of interventions. The uptake of any practice should depend on both farmers' characteristics and factors inherent in the practice itself. This paper, therefore, examines procedures for incorporating the applicability of CSA practices in a farm-level analysis based on the investigations conducted in King Cetshwayo District Municipality (KCDM) of the KwaZulu-Natal (KZN) Province of South Africa. How the farmers perceived the social, technical, economic, and environmental compatibility of the practices constituted the key goal of the inquiry. Data were collected through structured interviews using close-ended questionnaires, from a sample of 327 small-scale farmers (farmers with farm sizes of less than or equal to 5 hectares). The analysis made use of the Acceptance Level Index (*ALI*) and Composite Score Index (CSI). This paper establishes that, based on social compatibility, the farmers showed high acceptance for cultivation of cover crops (*ALI* = 574), agroforestry (*ALI* = 559), and diet improvement for animals (*ALI* = 554), based on technical compatibility, the use of organic manure (*ALI* = 545), rotational cropping (*ALI* = 529), mulching (*ALI* = 525) and cultivation of cover crops (*ALI* = 533) were highly accepted. With economic compatibility in perspective, the farmers showed high preference for mulching (*ALI* = 541), organic manure (*ALI* = 542) and rotational cropping (*ALI* = 515), while the use of organic manure (*ALI* = 524) was highly embraced based on environmental compatibility. Consequently, it is recommended that policies aimed at mainstreaming CSA technologies should pay adequate attention to their applicability in locations under consideration and emphasize the critical role of the provision of information on CSA technologies or practices.

**Keywords:** food security; greenhouse gas mitigation; holistic compatibility; resilience improvement

## 1. Introduction

There is a rapidly growing interest in climate-smart agriculture (CSA), especially in the developing world as a result of its promising potential to improve food security, climate change resilience, and mitigation of greenhouse gas (GHG) emissions [1]. Climate-smart agriculture is crucial in African countries where the agricultural sector is highly vulnerable to changes in climatic conditions, and agricultural growth plays a significant role in economic development [2]. There have been efforts on CSA that have brought about different initiatives such as the Africa CSA alliance and similar programs

and concepts. However, there is still the need for methodologies and approaches that will factor in the comprehensive combination of socioeconomic and biophysical realities to mainstream CSA technologies [3].

Mainstreaming CSA depends on institutional instruments, resource tenancy, socioeconomic factors, and climate and ecology setting [4]. These influencers are critical players in the acceptance of CSA practices at the farm-level [4]. For example, in low-input small-scale farming systems, increased productivity, and adaptive capacity will be prioritized over increasing carbon sequestration and emission reduction [5]. Farmers may be more interested in the applicability and immediate benefits that will accrue from CSA adoption than the long-term technical benefits it promises. Applicability of CSA technologies and practices in the context of this paper addresses the suitability of the technologies and practices in relation to the prevailing societal and biophysical conditions of the location under consideration. Farmers' decisions on whether and how to adapt agricultural technologies are influenced by the dynamic interaction between the characteristics of the technologies and an array of conditions and circumstances [6].

The characteristics of agricultural technology play significant roles in technology adaptation and implementation [7]. Mignouna et al. [8] argued that farmers who perceive a technology to be applicable and compatible to their needs and environment are likely to regard such technology as a positive investment; hence, they are open to adaptation and implementation of such technology. The applicability of the technologies gets revealed in their acceptance and prioritization by farmers. However, since the adaptation and mitigation benefits of CSA are complementary and can sometimes be mutually reinforcing, achieving the triple-win effect of CSA include increasing productivity, adapting to climate change (resilience) and GHG mitigation [3]. The implementation of CSA technologies and practices can bring about the reduction of the impacts of climate change on agriculture [9]. Different studies suggest that mainstreaming CSA in the farming system can boost yields, enhance the efficiency of use of input, increase income from production and reduce GHG emissions [10–12].

The main difficulties for mainstreaming CSA in different agroecological zones are identifying and prioritizing the applicability of CSA practices, putting into consideration the risks from local climatic conditions and the need for such innovations [9]. To identify and comprehend the applicability of CSA technologies facilitates the planning and design of frameworks and structures meant to assist farmers in adapting against climate change and improving their resilience [9]. There is a need for the consideration of adaptation practices that have been adequately tested and accepted by farmers concerning location-specific climate-related risks when making efforts to mainstream CSA [13].

Because farming systems in Africa are complex, research is critical to inform and support adaptation decision making. This support includes areas that will help the agricultural transformation needed and being advocated for in African agricultural system. In mainstreaming CSA, there is the need to systematically harness the limited available resources to augment the triple benefits of CSA. There is, therefore, the need for comprehensive information on the identification and prioritization of locally appropriate CSA practices and the enabling environment needed for the adaptation and sustenance of the uptake.

Despite the urgency of understanding the applicability of CSA at the farm level, many CSA programs are deficient of information needed for successful CSA implementation among farmers [9,14]. Information and evidence on the applicability of CSA practices or technologies, as well as farmers' approval and prioritization (particularly the local farmers) can assist stakeholders in making strategic decisions that will enhance government policies and institutional arrangements to achieve desired results. Given the relevance of the information on the applicability of CSA practices, this study examines the applicability of CSA practices in light of how farmers perceive the social, technical, economic, and environmental compatibility of the practices. This paper considered the CSA practices identified from the sample farmers in King Cetshwayo District Municipality (KCDM), covering a wide range of agronomic and animal husbandry practices, land and enterprise management regimes, and resource use levels as detailed in Table 1.

**Table 1.** Farmers' Perception of Social Compatibility of Climate-Smart Agricultural Practices.

| Climate-Smart Agricultural Practice | Level of Social Acceptance | | | | | | | | | | | | | | |
|---|---|---|---|---|---|---|---|---|---|---|---|---|---|---|---|
| | Mthonjaneni | | | | | uMhlathuze | | | | | Combined Analysis | | | | |
| | A No (%) | N No (%) | NA No (%) | *ALI* | CA | A No (%) | N No (%) | NA No (%) | *ALI* | CA | A No (%) | N No (%) | NA No (%) | *ALI* | CA |
| Planting of Cover Crops | 84 (77.1) | 25 (22.9) | 0 (0.0) | 193 | High | 163 (74.7) | 55 (25.2) | 0 (0.0) | 381 | High | 247 (75.5) | 80 (24.5) | 0 (0.0) | 574 | High |
| Agroforestry | 85 (78.0) | 18 (16.6) | 6 (5.5) | 190 | High | 171 (78.5) | 29 (13.3) | 18 (8.3) | 371 | High | 256 (78.3) | 47 (14.4) | 24 (7.3) | 559 | High |
| Crop Rotation | 80 (73.4) | 29 (26.6) | 0 (0.0) | 189 | Medium | 141 (64.7) | 61 (28.0) | 16 (7.3) | 343 | Medium | 221 (67.6) | 90 (27.5) | 16 (4.9) | 532 | Medium |
| Mulching | 87 (79.8) | 15 (13.8) | 7 (6.4) | 189 | Medium | 153 (70.2) | 48 (22.0) | (17 (7.8) | 354 | Medium | 240 (73.4) | 63 (19.3) | 24 (7.3) | 543 | Medium |
| Use of Organic Manure | 81 (74.3) | 26 (23.9) | 2 (1.8) | 188 | Medium | 134 (61.4) | 66 (30.3) | 18 (8.3) | 334 | Medium | 215 (65.8) | 92 (28.1) | 20 (6.1) | 522 | Medium |
| Efficient Manure Management | 78 (71.5) | 31 (28.4) | 0 (0.0) | 187 | Medium | 141 (64.7) | 61 (28.0) | 16 (7.3) | 343 | Medium | 219 (67.0) | 92 (28.1) | 16 (4.9) | 530 | Medium |
| Integrated Crop-Livestock Management | 77 (70.6) | 31 (28.4) | 1 (0.9) | 185 | Medium | 148 (67.9) | 43 (19.7) | 27 (12.4) | 339 | Medium | 225 (68.8) | 74 (22.6) | 28 (8.6) | 524 | Medium |
| Crop Diversification | 79 (72.5) | 25 (22.9) | 5 (4.6) | 183 | Medium | 137 (62.8) | 53 (24.3) | 28 (12.8) | 327 | Medium | 216 (66.1) | 78 (23.9) | 33 (10.1) | 510 | Medium |
| Planting of Drought- and heat-tolerant Crops | 75 (68.8) | 31 (28.4) | 3 (2.8) | 181 | Medium | 128 (58.7) | 58 (26.6) | 32 (14.7) | 314 | Low | 203 (62.1) | 89 (27.2) | 35 (10.7) | 495 | Low |
| Conservation Agriculture | 77 (70.6) | 23 (21.1) | 9 (8.3) | 177 | Medium | 135 (61.9) | 50 (22.9) | 33 (15.1) | 320 | Low | 212 (64.8) | 73 (22.3) | 42 (12.8) | 497 | Low |
| Diet Improvement for Animals | 78 (71.5) | 20 (18.3) | 11 (10.1) | 176 | Medium | 162 (74.3) | 54 (24.8) | 2 (0.9) | 378 | High | 240 (73.4) | 74 (22.6) | 13 (4.0) | 554 | High |
| Improved Grazing | 75 (68.8) | 25 (22.9) | 9 (8.3) | 175 | Medium | 140 (64.2) | 67 (30.7) | 11 (5.0) | 347 | Medium | 215 (65.8) | 92 (28.1) | 20 (6.1) | 522 | Medium |
| Use of Wetlands | 76 (69.7) | 21 (19.3) | 12 (11.0) | 173 | Low | 136 (62.4) | 46 (21.2) | 36 (16.5) | 318 | Low | 212 (64.8) | 67 (20.5) | 48 (14.7) | 491 | Low |
| Soil Conservation | 74 (67.9) | 20 (18.3) | 15 (13.8) | 168 | Low | 147 (67.4) | 38 (17.4) | 33 (15.1) | 332 | Medium | 221 (67.6) | 58 (17.7) | 48 (14.7) | 500 | Low |

Source: Survey Data (2018/19). Notes: A = Acceptable; N = Neutral; NA = Not Acceptable; *ALI* = Acceptance Level Index; CA = Category of Acceptance.

## 2. Materials and Methods

### 2.1. Selection and Socioeconomic Profile of the Study Area

This study was conducted in KCDM of the KwaZulu-Natal (KZN) Province of South Africa. The KZN Province stretches from the Indian Ocean in the east, to the Drakensberg Mountains in the west, where Lesotho is the neighboring country. The province has one metropolitan municipality (eThekwini Metropolitan Municipality) and ten district municipalities, which are broken down into 43 local municipalities. The KZN Province has a land area of 94,361 km$^2$, which constitutes 7.7% of South Africa's landmass [15]. The selection of the study area was carried out with the aid of a multistage sampling technique. The province was purposively selected because it ranks as the most important agricultural area on the bases of the size of the farming households and agricultural production levels [16]. The district municipality (one out of ten) was selected randomly, and two local municipalities out of five (Mthonjaneni and uMhlathuze) were purposefully selected based on their agricultural potential. Figure 1 is a map showing KCDM and its local municipalities.

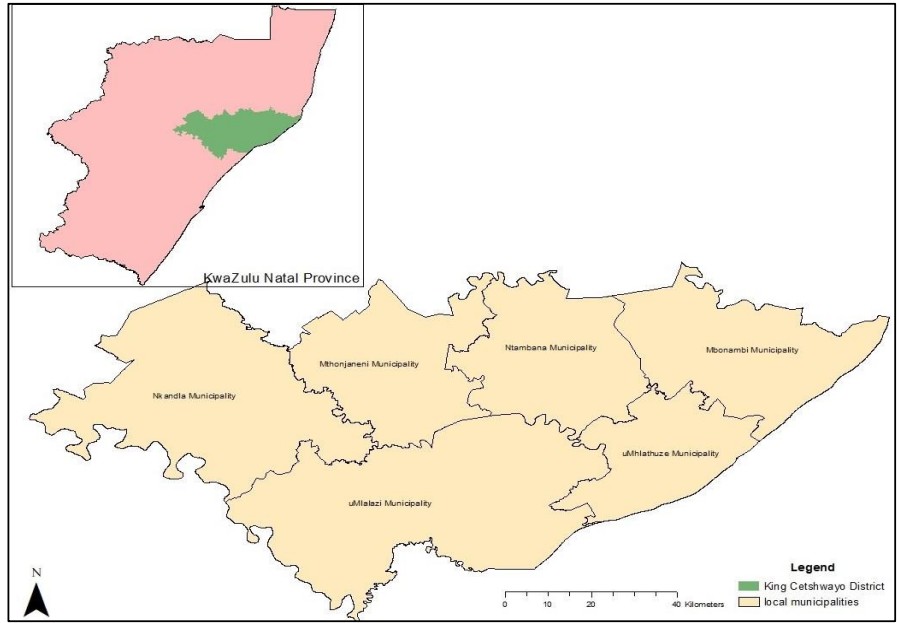

**Figure 1.** Map showing King Cetshwayo District Municipality and its local municipalities.

### 2.2. Research Design

This study combines a cross-sectional research design with a quantitative research approach. A cross-sectional research design is used to find the correlation between variables at a specific point in time [17]. It can engage data from different disciplines and contrasting observational studies [18]. A cross-sectional research design is used to analyze and draw inferences from the differences existing between people, subjects or phenomena [18]. A cross-sectional research design was considered appropriate for this study because it allows for the use of survey method for data collection at a particular point in time, and it enables rational and lucid conclusions, despite being relatively more cost and time-saving [17,18]. The rationale behind the choice of a cross-sectional research design for this study is to be able to get a deeper insight into the applicability of CSA technology in the small-scale farming system, despite resource constraints.

Conceptual Framework

This paper conceptualizes the applicability of CSA practices through four main components; social, technical, economic, and environmental compatibility. Agricultural technology implementation

is influenced by much more than the socioeconomic characteristics of the farmers. Despite past studies identifying the determinants of technology adoption or adaptation in agriculture [7,19–21], there are still variations which can only be explained by the exogenous factors that were not captured by models of past studies focusing on characteristics of farmers when analyzing the determinants of adoption or technology transfer. This line of thinking deviates from the conventional approach that focuses on the inherent characteristics of the technologies or practices which could be of significant influence on adoption or transfer [3].

Technology intervention or introduction does not necessarily result in automatic adoption or transfer. There are key elements involved in the transfer or adoption of technologies. Farmers' characteristics and the attributes inherent in technologies or practices are important elements actively involved in the dynamics of technology transfer or adoption, particularly in the small-scale farming system [3,7,21]. The tendency of CSA to be content and location-specific makes the adoption of CSA practices to be more influence-prone [4,13]. The location (where) of intervention determines the technological requirements (what) at play, which in turn determines the CSA practices that could be implemented. However, in adapting the available practices, potential adapters will consider the attributes of the CSA technologies or practices with how they apply to different conditions in terms of social, technical, economic, and environmental compatibility.

Technology adoption addresses decisions regarding new technologies, while technology adaptation addresses decisions regarding existing technologies. This paper focuses on existing techniques in the study area/s. The social compatibility of the CSA packages addresses how the packages are well-suited with the culture, values, and norms of the location in consideration. The technical compatibility addresses how easy it is for the packages to be adapted and successfully implemented. The economic compatibility addresses the financial implication of implementation, while the environmental compatibility addresses the likely effect of implementation on the agricultural system and the environment. The conceptual framework in Figure 2 illustrates the influence of applicability indicators on CSA implementation.

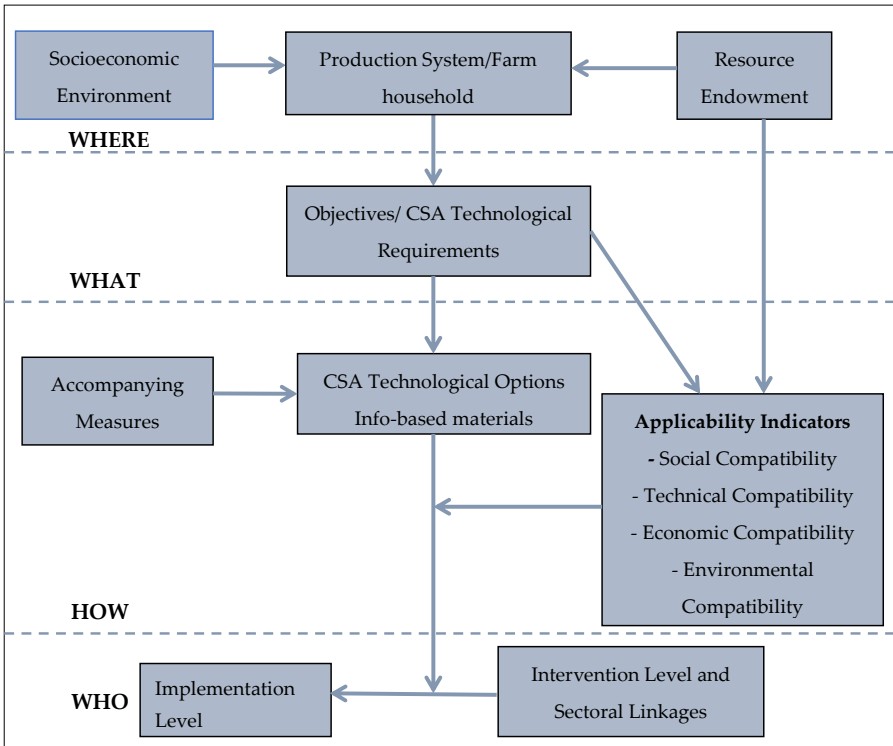

**Figure 2.** Conceptual framework – location-specific assessment concept of the applicability of climate-smart agriculture (CSA).

### 2.3. Study Population and Sampling Procedure

The study targeted small-scale farming households in KCDM. The small-scale farming households are farming households with farm sizes of less than or equal to 5 hectares [22]. Considerations were given to both crop and livestock farming households. As earlier stated, Mthonjaneni and uMhlathuze Local Municipalities were selected, from the district, because of their agricultural potential. Villages in the selected municipalities were approached for sampling based on the information from the Department of Agriculture and Rural Development (DARD). According to DARD records, the number of small-scale farming households in uMhlathuze and Mthonjaneni stood at 1440 and 720, respectively, at the time of the study, totalling 2160 in the two municipalities. A sample size calculator was used to obtain the total sample size of 327 based on a 95 per cent confidence level and confidence interval of 5. Based on the method of Hoyle et al. [23], the sample size for each local municipality was computed as follows:

- Sample size for uMhlathuze – n = (1440/2160) * 327 = 218
- Sample size for Mthonjaneni – n = (720/2160) * 327 = 109

A random selection of the calculated number of small-scale farming households was conducted in the villages in each of the local municipalities. A random selection allowed for an equal chance of selection for every small-scale farming household in the selected local municipalities.

### 2.4. Data Collection

Prior to data collection, ethical clearance was obtained. The study met the specified standards set in terms of research and safety, as stated in the university's policies and procedures on the research ethics document. Since the study involved human participation, the study took into cognizance the respect of the dignity of respondents in the research process. Municipal authorities were also consulted and carried along through the course of the study. Data were collected through structured interviews with the use of pre-tested close-ended questionnaires. The structured interview elicited relevant information needed for empirical findings from the study. The questionnaires were pre-tested on 35 respondents before being finalized. According to Moore et al. [24] and Connelly [25], 10 percent of the actual sample size can be used as the size needed for a pilot study. The questionnaires were pre-tested to test for the validity and reliability of the questionnaires and therefore ensure the validity and reliability of the data collected. The participants of the pilot study were excluded from the actual study. Data collection was carried out in the season preceding the survey period to ensure a uniform and complete dataset. The questionnaires were interviewer-administered to avoid misinterpretation or misunderstanding of questions. Data were collected at the household level on a face-to-face basis with the actual respondents being the person(s) responsible for household farming activities. Engaging the actual person(s) responsible for the household farming activities facilitated eliciting a robust response from the households. Data collection was carried out between August 2018 and January 2019 within the working hours of 08h00 to 16h00. Periods of social functions such as funerals and weddings, as well as days of social grant collection were not included for data collection.

### 2.5. Data Analysis

The applicability of the identified CSA practices in the small-scale farming system was assessed using the acceptance level index (*ALI*) and the composite score index. The *ALI* was adapted from the adaptation strategy use index [26–28], and was used to analyze the level of acceptance of the identified CSA practices. Respondents assessed CSA practices by using a three-point rating scale 2, 1, and 0 to denote acceptable, neutral, and not acceptable, respectively.

The following index formula formed the basis for the computation of the relative level of acceptance of the practices:

$$ALI = AL_{nt} \, X \, 0 + \, AL_n X \, 1 + \, AL_a X \, 2 \tag{1}$$

where;

$ALI$ = Acceptance Level Index

$AL_{nt}$ = Frequency of farming households who reckoned the climate-smart agricultural practices under consideration as not acceptable

$AL_n$ = Frequency of farming households who were indifferent about the climate-smart agricultural practices under consideration

$AL_a$ = Frequency of farming households who reckoned the climate-smart agricultural practices under consideration as acceptable

With the responses from the farmers, the maximum points for the $ALI$ for each CSA practice can only be 218 ($ALI = 109 \times 2$), 436 ($ALI = 218 \times 2$) and 654 ($ALI = 327 \times 2$) in the two local municipalities and districts, respectively and 0 as minimum point. A composite score was used to generate three categories of acceptance levels, namely high, medium and low [17].

Where:

High category = $ALI$ that falls between the maximum and (Mean + S.D) points

Medium category = $ALI$ that falls between the upper and lower categories

Low category = $ALI$ that falls between (Mean − S.D) and 0

## 3. Results

### 3.1. Farmers' Perception of Social Compatibility of Climate-Smart Agricultural Practices

Table 1 presents the results. Given the analysis of the acceptance level index of the practices, the results reveal that the planting of cover crops ($ALI = 193$) and agroforestry ($ALI = 190$) had a high level of social acceptance among the farmers in Mthonjaneni Municipality, while agroforestry ($ALI = 371$), cultivation of cover crops (381) and diet improvement for animals ($ALI = 378$) had a high level of social acceptance among the farmers in uMhlathuze Municipality. Results from the combined (Mthonjaneni and uMhlathuze Municipality) analysis show a similar result obtained for uMhlathuze Municipality, where agroforestry ($ALI = 559$), cultivation of cover crops ($ALI = 574$) and diet improvement for animals ($ALI = 554$) had a high level of social acceptance among the farmers in KCDM.

Farmers in Mthonjaneni Municipality showed a low social acceptance for livestock diet improvement ($ALI = 196$), the use of wetland ($ALI = 194$) and soil conservation ($ALI = 188$), while the farmers in uMhlathuze Municipality showed low social acceptance for conservation agriculture ($ALI = 320$), use of wetland ($ALI = 318$) and cultivation of drought- and heat-tolerant crops ($ALI = 314$). For the whole sample, soil conservation ($ALI = 500$), cultivation of crops with high drought- and heat-tolerance ($ALI = 495$), conservation agriculture ($ALI = 497$) and use of wetland ($ALI = 491$) had low social acceptance among the farmers in KCDM. The other identified CSA practices had a medium level of social acceptance among farmers (Table 1).

### 3.2. Farmers' Perception of Technical Compatibility of Climate-Smart Agricultural Practices

Table 2 reveals that the results on the level of farmers' acceptance of the identified CSA practices based on the ease of adoption or use of those practices. Results show that the use of organic manure ($ALI = 179$), rotational cropping ($ALI = 179$), crop diversification ($ALI = 175$) and mulching ($ALI = 174$) had a high level of acceptance among the farmers in Mthonjaneni Municipality, while the use of organic manure ($ALI = 366$), cultivation of cover crops ($ALI = 364$), mulching ($ALI = 351$) and rotational cropping ($ALI = 350$) had a high level of acceptance among the farmers in uMhlathuze Municipality. Results from the combined analysis show that the use of organic manure ($ALI = 545$), cultivation of cover crops ($ALI = 533$), crop rotation ($ALI = 529$), mulching ($ALI = 525$) and had a high level of acceptance among the farmers in KCDM.

**Table 2.** Farmers' Perception of Technical Compatibility of Climate-Smart Agricultural Practices.

| Climate-Smart Agricultural Practice | Level of Acceptance based on Technicality | | | | | | | | | | | | | | |
| --- | --- | --- | --- | --- | --- | --- | --- | --- | --- | --- | --- | --- | --- | --- |
| | Mthonjaneni | | | | | uMhlathuze | | | | | Combined Analysis | | | | |
| | A No (%) | N No (%) | NA No (%) | *ALI* | CA | A No (%) | N No (%) | NA No (%) | *ALI* | CA | A No (%) | N No (%) | NA No (%) | *ALI* | CA |
| Use of Organic Manure | 85 (75.2) | 9 (8.3) | 15 (13.8) | 179 | High | 168 (77.1) | 30 (13.8) | 20 (9.2) | 366 | High | 253 (77.4) | 39 (11.9) | 35 (10.7) | 545 | High |
| Crop Rotation | 82 (75.2) | 15 (13.8) | 12 (11.0) | 179 | High | 155 (71.1) | 40 (18.4) | 23 (10.6) | 350 | High | 237 (72.5) | 55 (16.8) | 35 (10.7) | 529 | High |
| Crop Diversification | 76 (69.7) | 23 (21.1) | 10 (9.2) | 175 | High | 141 (64.7) | 49 (22.5) | 28 (12.8) | 331 | Medium | 217 (66.4) | 72 (22.0) | 38 (11.6) | 506 | Medium |
| Mulching | 83 (76.2) | 8 (7.3) | 18 (16.5) | 174 | High | 158 (72.5) | 35 (16.1) | 25 (11.5) | 351 | High | 241 (73.7) | 43 (13.2) | 43 (13.2) | 525 | High |
| Planting of Cover Crops | 80 (73.4) | 9 (8.3) | 20 (18.4) | 169 | Medium | 172 (78.9) | 20 (9.2) | 26 (11.9) | 364 | High | 252 (77.1) | 29 (8.9) | 46 (14.1) | 533 | High |
| Use of Wetlands | 70 (64.2) | 17 (15.6) | 22 (20.2) | 157 | Medium | 132 (60.6) | 46 (21.1) | 40 (18.4) | 310 | Medium | 202 (61.8) | 63 (19.3) | 62 (19.0) | 467 | Medium |
| Integrated Crop-Livestock Management | 73 (67.0) | 11 (10.1) | 25 (22.9) | 157 | Medium | 150 (68.8) | 23 (10.6) | 45 (20.6) | 323 | Medium | 223 (68.2) | 34 (10.4) | 70 (21.4) | 480 | Medium |
| Improved Grazing | 77 (70.6) | 2 (1.8) | 30 (27.5) | 156 | Medium | 146 (67.0) | 39 (17.9) | 33 (15.1) | 331 | Medium | 223 (68.2) | 41 (12.6) | 63 (19.3) | 487 | Medium |
| Planting of Drought- and Heat-Tolerant Crops | 68 (62.4) | 6 (5.5) | 35 (32.1) | 142 | Medium | 130 (59.6) | 32 (14.7) | 56 (25.7) | 292 | Low | 198 (60.6) | 38 (11.6) | 91 (27.8) | 434 | Low |
| Efficient Manure Management | 67 (61.5) | 6 (5.5) | 36 (33.0) | 140 | Medium | 141 (64.7) | 42 (19.3) | 35 (16.1) | 324 | Medium | 208 (63.6) | 48 (14.7) | 71 (21.7) | 464 | Medium |
| Conservation Agriculture | 60 (55.1) | 19 (17.4) | 30 (27.5) | 139 | Low | 120 (55.1) | 53 (24.3) | 45 (20.6) | 293 | Low | 180 (55.1) | 72 (22.0) | 75 (23.0) | 432 | Low |
| Agroforestry | 65 (59.6) | 9 (8.3) | 35 (32.1) | 139 | Low | 133 (61.0) | 25 (11.5) | 60 (27.5) | 291 | Low | 198 (60.6) | 34 (10.4) | 95 (29.1) | 430 | Low |
| Soil Conservation | 67 (61.5) | 5 (4.6) | 37 (33.9) | 139 | Low | 129 (59.2) | 26 (11.9) | 63 (28.9) | 284 | Low | 196 (59.9) | 31 (9.5) | 100 (30.6) | 423 | Low |
| Diet Improvement for Animals | 65 (59.6) | 6 (5.5) | 38 (34.9) | 136 | Low | 138 (63.3) | 31 (14.2) | 49 (22.5) | 307 | Medium | 203 (62.1) | 37 (11.3) | 87 (26.6) | 443 | Medium |

Source: Survey Data (2018/19). Notes: A = Acceptable; N = Neutral; NA = Not Acceptable; *ALI* = Acceptance Level Index; CA = Category of Acceptance.

Results in Table 2 further reveal that farmers in Mthonjaneni Municipality unveiled a low level of acceptance for conservation agriculture (*ALI* = 139), agroforestry (*ALI* = 139), soil conservation (*ALI* = 139) and diet improvement for animals (*ALI* = 136) based on a technicality, while their counterparts in uMhlathuze Municipality laid out a low level of acceptance for conservation agriculture (*ALI* = 293), cultivation of drought- and heat-tolerant crops (*ALI* = 292), agroforestry (*ALI* = 291), and soil conservation (*ALI* = 284). Results from the combined analysis show that based on the technicality of the CSA practices, farmers in KCDM showed a low level of acceptance for cultivation of drought and heat-tolerant crop (*ALI* = 434), conservation agriculture (*ALI* = 432), agroforestry (*ALI* = 430), and soil conservation (*ALI* = 423).

### 3.3. Farmers' Perception of Economic Compatibility of Climate-Smart Agricultural Practices

Table 3 shows the results obtained on the level of farmers' acceptance of the identified CSA practices, with the economics of the use of the practices in perspective. Results show that rotational cropping (*ALI* = 180) and mulching (*ALI* = 178) were highly accepted by the farmers in Mthonjaneni Municipality, with the perception of being economical, while farmers in uMhlathuze Municipality highly accepted the use of organic manure (*ALI* = 361), mulching and cultivation of cover crops (*ALI* = 351) based on the economics of use of those practices. Results from the combined analysis reveal that the use of organic manure (*ALI* = 542), mulching (*ALI* = 541) and rotational cropping (*ALI* = 515) had a high level of acceptance among the farmers in KCDM based on the economic compatibility.

Table 3 further shows that the farmers in Mthonjaneni Municipality showed a low level of acceptance for the cultivation of drought- and heat-tolerant crops (*ALI* = 135) and diet improvement for animals (*ALI* = 135) based on economics of use, while their counterparts in uMhlathuze Municipality had a low level of acceptance for cultivation of drought- and heat-tolerant crops (*ALI* = 247) and agroforestry (*ALI* = 235). Results from the combined analysis show that with the economics of the use of the practices in perspective, the farmers in KCDM had a low level of acceptance for agroforestry (*ALI* = 436) and cultivation of drought- and heat-tolerant crops (*ALI* = 430).

### 3.4. Farmers' Perception of Environmental Compatibility of Climate-Smart Agricultural Practices

Table 4 shows the level of farmers' acceptance of the identified CSA practices based on the perception of the environmental friendliness of the practices. Results show that farmers in Mthonjaneni Municipality showed a high level of acceptance for agroforestry (*ALI* = 178) and rotational cropping (*ALI* = 175), with environmental compatibility in perspective, while it was the use of organic manure (*ALI* = 358) that was highly accepted by the farmers in uMhlathuze Municipality based on environmental compatibility. Results from the combined analysis reveal that farmers in KCDM highly accepted the use of organic manure (*ALI* =524) based on their perspective on the environmental friendliness of the identified practices. The remaining identified CSA practices had a medium level of acceptance among the farmers.

Farmers in Mthonjaneni Municipality showed a relatively low level of acceptance for the use of wetland (*ALI* = 159), cultivation of drought-tolerant crops (*ALI* = 158), efficient manure management (*ALI* = 158) and diet improvement for animals (*ALI* = 158) based on their understanding of the environmental friendliness of the practices. However, the farmers in uMhlathuze Municipality showed a relatively low level of acceptance for crop diversification (*ALI* = 296) and the use of wetland (*ALI* = 286) based on their understanding of the environmental friendliness of these practices. Results from the combined analysis show that the use of wetland (*ALI* = 445) and efficient manure management (*ALI* = 436) had a low acceptance from the farmers in KCDM, with environmental compatibility in perspective (Table 4).

**Table 3.** Farmers' Perception of Economic Compatibility of Climate-Smart Agricultural Practices.

| Climate-Smart Agricultural Practice | Level of Acceptance based on Economics of Use | | | | | | | | | | | | | | |
|---|---|---|---|---|---|---|---|---|---|---|---|---|---|---|
| | Mthonjaneni | | | | | uMhlathuze | | | | | Combined Analysis | | | | |
| | A No (%) | N No (%) | NA No (%) | *ALI* | CA | A No (%) | N No (%) | NA No (%) | *ALI* | CA | A No (%) | N No (%) | NA No (%) | *ALI* | CA |
| Crop Rotation | 83 (76.2) | 14 (12.8) | 12 (11.0) | 180 | High | 151 (69.3) | 34 (15.6) | 33 (15.1) | 336 | Medium | 234 (71.6) | 47 (14.4) | 45 (13.8) | 515 | High |
| Mulching | 80 (73.4) | 18 (16.5) | 11 (10.1) | 178 | High | 168 (77.1) | 23 (10.6) | 27 (12.4) | 359 | High | 248 (75.8) | 45 (13.8) | 38 (11.6) | 541 | High |
| Use of Organic Manure | 75 (68.8) | 16 (14.7) | 18 (16.5) | 166 | Medium | 173 (79.4) | 15 (6.9) | 30 (13.8) | 361 | High | 248 (75.8) | 46 (14.1) | 48 (14.7) | 542 | High |
| Improved Grazing | 77 (70.6) | 12 (11.0) | 20 (18.4) | 166 | Medium | 144 (66.1) | 36 (16.5) | 38 (17.4) | 324 | Medium | 221 (67.6) | 50 (15.3) | 58 (17.7) | 492 | Medium |
| Crop Diversification | 72 (66.1) | 17 (15.6) | 20 (18.4) | 161 | Medium | 134 (61.5) | 48 (22.0) | 36 (16.5) | 316 | Medium | 206 (63.0) | 53 (16.2) | 56 (17.1) | 465 | Medium |
| Conservation Agriculture | 70 (64.2) | 14 (12.8) | 25 (22. 9) | 154 | Medium | 130 (59.6) | 18 (8.3) | 70 (32.1) | 278 | Medium | 200 (61.2) | 84 (25.7) | 95 (29.1) | 484 | Medium |
| Planting of Cover Crops | 68 (62.4) | 16 (14.7) | 25 (22.9) | 152 | Medium | 156 (71.6) | 39 (17.9) | 23 (10.6) | 351 | High | 224 (68.5) | 39 (11.9) | 48 (14.7) | 487 | Medium |
| Agroforestry | 60 (55.1) | 27 (24.8) | 22 (20.2) | 147 | Medium | 102 (46.8) | 31 (14.2) | 85 (39.0) | 235 | Low | 162 (49.5) | 112 (34.3) | 107 (32.7) | 436 | Low |
| Integrated Crop-Livestock Management | 65 (59.6) | 17 (15.6) | 27 (24.8) | 147 | Medium | 146 (67.0) | 24 (11.0) | 48 (22.0) | 316 | Medium | 211 (64.5) | 65 (19.9) | 75 (22.9) | 487 | Medium |
| Efficient Manure Management | 65 (59.6) | 16 (14.7) | 28 (25.7) | 146 | Medium | 138 (63.3) | 47 (21.6) | 33 (15.1) | 323 | Medium | 203 (62.1) | 49 (15.0) | 61 (18.7) | 455 | Medium |
| Use of Wetlands | 65 (59.6) | 16 (14.7) | 28 (25.7) | 146 | Medium | 121 (55.5) | 29 (13.3) | 68 (31.2) | 271 | Medium | 186 (56.9) | 84 (25.7) | 96 (29.4) | 456 | Medium |
| Soil Conservation | 65 (59.6) | 11 (10.1) | 33 (30.3) | 141 | Medium | 125 (57.3) | 28 (12.8) | 65 (29.8) | 278 | Medium | 190 (58.1) | 76 (23.2) | 98 (30.0) | 456 | Medium |
| Planting of Drought- and Heat-Tolerant Crops | 62 (56.9) | 11 (10.1) | 36 (33.0) | 135 | Low | 108 (49.5) | 31 (14.2) | 79 (36.2) | 247 | Low | 170 (52.0) | 90 (27.5) | 115 (35.2) | 430 | Low |
| Diet Improvement for Animals | 62 (56.9) | 11 (10.1) | 36 (33.0) | 135 | Low | 129 (59.2) | 35 (16.1) | 54 (24.8) | 293 | Medium | 191 (58.4) | 65 (19.9) | 90 (27.5) | 447 | Medium |

Source: Survey Data (2018/19). Notes: A = Acceptable; N = Neutral; NA = Not Acceptable; *ALI* = Acceptance Level Index; CA = Category of Acceptance.

**Table 4.** Farmers' Perception of Environmental Compatibility of Climate-Smart Agricultural Practices.

| Climate-Smart Agricultural Practice | Level of Acceptance based on Environmental Friendliness | | | | | | | | | | | | | | |
| --- | --- | --- | --- | --- | --- | --- | --- | --- | --- | --- | --- | --- | --- | --- | --- |
| | Mthonjaneni | | | | | uMhlathuze | | | | | Combined Analysis | | | | |
| | A No (%) | N No (%) | NA No (%) | *ALI* | CA | A No (%) | N No (%) | NA No (%) | *ALI* | CA | A No (%) | N No (%) | NA No (%) | *ALI* | CA |
| Agroforestry | 80 (73.4) | 18 (16.5) | 11 (10.1) | 178 | High | 128 (58.7) | 55 (25.2) | 35 (16.1) | 311 | Medium | 208 (63.6) | 73 (22.3) | 46 (14.1) | 489 | Medium |
| Crop Rotation | 75 (68.8) | 25 (22.9) | 9 (8.3) | 175 | High | 146 (67.0) | 37 (17.0) | 35 (16.1) | 329 | Medium | 221 (67.6) | 62 (19.0) | 44 (13.5) | 504 | Medium |
| Conservation Agriculture | 70 (64.2) | 28 (25.7) | 11 (10.1) | 168 | Medium | 135 (61.9) | 60 (27.5) | 23 (10.6) | 330 | Medium | 205 (62.7) | 88 (26.9) | 34 (10.4) | 498 | Medium |
| Planting of Cover Crops | 70 (64.2) | 27 (24.8) | 12 (11.0) | 167 | Medium | 143 (65.6) | 25 (11.5) | 50 (22.9) | 311 | Medium | 213 (65.1) | 52 (15.9) | 62 (19.0) | 478 | Medium |
| Use of Organic Manure | 73 (67.0) | 20 (18.3) | 16 (14.7) | 166 | Medium | 160 (73.4) | 38 (17.4) | 20 (9.2) | 358 | High | 233 (71.3) | 58 (17.7) | 36 (11.0) | 524 | High |
| Crop Diversification | 68 (62.4) | 30 (27.5) | 11 (10.1) | 166 | Medium | 128 (58.7) | 40 (18.4) | 50 (22.9) | 296 | Low | 196 (59.9) | 70 (21.4) | 61 (18.6) | 462 | Medium |
| Mulching | 67 (61.5) | 32 (29.4) | 10 (9.2) | 166 | Medium | 140 (64.2) | 30 (13.8) | 48 (22.0) | 310 | Medium | 207 (63.3) | 62 (19.0) | 58 (17.7) | 476 | Medium |
| Soil Conservation | 66 (60.6) | 33 (30.3) | 10 (9.2) | 165 | Medium | 138 (63.3) | 62 (28.4) | 18 (8.3) | 338 | Medium | 204 (62.4) | 95 (29.1) | 28 (8.6) | 503 | Medium |
| Integrated Crop-Livestock Management | 65 (59.6) | 35 (32.1) | 9 (11.0) | 165 | Medium | 140 (64.2) | 50 (22.9) | 28 (12.8) | 328 | Medium | 205 (62.7) | 85 (26.0) | 37 (11.3) | 495 | Medium |
| Improved Grazing | 65 (59.6) | 32 (29.4) | 12 (11.0) | 162 | Medium | 135 (61.9) | 40 (18.4) | 43 (19.7) | 320 | Medium | 200 (61.2) | 72 (22.0) | 55 (16.8) | 472 | Medium |
| Use of Wetlands | 62 (56.9) | 35 (32.1) | 12 (11.0) | 159 | Low | 113 (51.8) | 60 (27.5) | 45 (20.6) | 286 | Low | 175 (53.5) | 95 (29.1) | 57 (17.7) | 445 | Low |
| Planting of Drought- and Heat-Tolerant Crops | 60 (55.1) | 38 (34.9) | 11 (10.1) | 158 | Low | 135 (61.9) | 66 (30.3) | 17 (7.8) | 336 | Medium | 195 (59.6) | 104 (31.8) | 28 (8.6) | 494 | Medium |
| Efficient Manure Management | 60 (55.1) | 38 (34.9) | 11 (10.1) | 158 | Low | 120 (55.1) | 38 (17.4) | 60 (27.5) | 320 | Medium | 180 (55.1) | 76 (23.2) | 71 (21.7) | 436 | Low |
| Diet Improvement for Animals | 61 (56.0) | 36 (33.0) | 12 (11.0) | 158 | Low | 136 (62.4) | 46 (21.1) | 36 (16.5) | 330 | Medium | 197 (60.2) | 82 (25.1) | 48 (14.7) | 476 | Medium |

Source: Survey Data (2018/19). Notes: A = Acceptable; N = Neutral; NA = Not Acceptable; *ALI* = Acceptance Level Index; CA = Category of Acceptance.

### 3.5. Farmers' Perception of Compatibility of Climate-Smart Agricultural Practices Across Domains

Analysis of the farmers' acceptance of the identified CSA practices across domains reveals those practices which rated well or had low acceptance across all or most of the social, technical, economic and environmental domains. Crop rotation was highly accepted across technical (*ALI* = 179), economic (*ALI* = 180) and environmental (*ALI* = 175) domains in Mthonjaneni Municipality. Mulching was highly accepted across technical (*ALI* = 174) and economic (*ALI* = 178) domains, while agroforestry was highly accepted across social (*ALI* = 190) and environmental (*ALI* = 178) domains in Mthonjaneni Municipality. In contrast, diet improvement for animals had low acceptance across technical (*ALI* = 136), economic (*ALI* = 135) and environmental (*ALI* = 158) domains in Mthonjaneni Municipality. Planting of drought- and heat-tolerant crops had low acceptance across economic (*ALI* = 135) and environmental (*ALI* = 158) domains. Soil conservation had low acceptance across social (*ALI* = 139) and technical (*ALI* = 168) domains, while the use of wetland had low acceptance across social (*ALI* = 173) and environmental (*ALI* = 159) domains.

The planting of cover crops was highly accepted across social (*ALI* = 381), technical (*ALI* = 364) and economic (*ALI* = 351) domains in uMhlathuze Municipality. The use of organic manure was highly accepted across technical (*ALI* = 366), economic (*ALI* = 361) and environmental (*ALI* = 358) domains, while mulching was highly accepted across technical (*ALI* = 351) and economic (*ALI* = 359) domains. On the other hand, planting of drought- and heat-tolerant crops had low acceptance across social (*ALI* = 314), technical (*ALI* = 292) and economic (*ALI* = 247) domains. Conservation agriculture had low acceptance across social (*ALI* = 320) and technical (*ALI* = 293) domains. The use of wetland had low acceptance across social (*ALI* = 318) and environmental (*ALI* = 286) domains, while agroforestry had low acceptance across technical (*ALI* = 291) and economic (*ALI* = 235) domains in uMhlathuze Municipality.

The use of organic manure was highly accepted across the technical (*ALI* = 545), economic (*ALI* = 542) and environmental (*ALI* = 524) domains in the combined analysis (KCDM). The planting of cover crops was highly accepted across social (*ALI* = 574) and technical (*ALI* = 533) domains, while mulching was highly accepted across technical (*ALI* = 525) and economic (*ALI* = 541) domains in the combined analysis (KCDM). In contrast, planting of drought- and heat-tolerant crops had low acceptance across social (*ALI* = 495), technical (*ALI* = 434) and economic (*ALI* = 430) domains in the combined analysis (KCDM). Conservation agriculture had low acceptance across social (*ALI* = 497) and technical (*ALI* = 432) domains. The use of wetland had low acceptance across social (*ALI* = 491) and environmental (*ALI* = 445) domains, while soil conservation had low acceptance across social (*ALI* = 500) and technical (*ALI* = 423) domains in the combined analysis (KCDM).

The results presented have shown how the sampled farmers perceived the CSA practices to be applicable based on the characteristics of the practices. Figure 3 is a schematic representation of the conditions under which the CSA practices are applicable in the small-scale farming system.

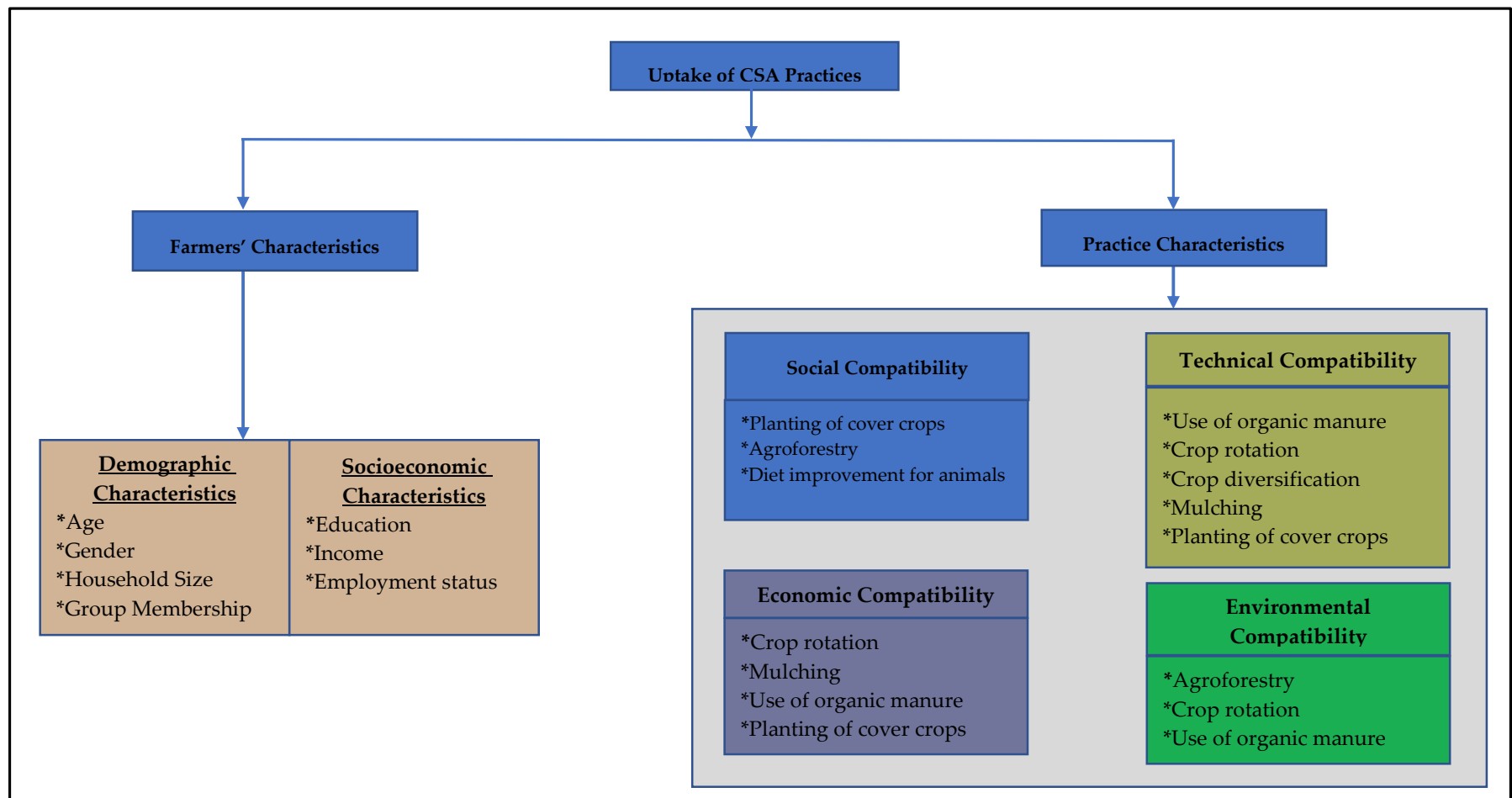

**Figure 3.** A schematic representation of the dynamics of the uptake of climate-smart agriculture (CSA) in the small-scale farming system.

## 4. Discussion

This paper provides insights into the applicability of CSA practices with a focus on how farmers perceive the social, technical, economic, and environmental compatibility of the practices. Past studies focus on the characteristics of farmers while researching the adaptation of agricultural technologies [28–30]. However, the focus on farmers' characteristics has not been holistic in addressing the determinants of technology adaptation [6]. This setting is reflected in the variations still observed in the adaptation of agricultural practices, which are only revealed in the exogenous factors not captured by models used in past studies.

The perception of farmers about the social compatibility of the identified CSA practices was informed by whether the practices conflicted with the cultural values of the farmers or gender-sensitive. Akudugu et al. [31] highlighted social and institutional factors as part of the factors influencing technology adoption and adaptation. Mignouna et al. [8] pointed out that belonging to a social group enhances social capital, which allows for trust and information exchange. Ojoko et al. [28] reveal that membership of a social group is a significant influencer of CSA adaptation. The strong social network among local farmers makes it difficult to implement techniques or technologies that do not properly fit into their social system. Eleven out of the fourteen identified practices showed either a high or medium level of acceptance by the farmers. This finding is expected as the practices considered in the study were identified among the farmers. Farmers highly embraced the planting of cover crops in both local municipalities. This finding suggests that the cultivation of cover crops had no conflict with the cultural values of the farmers, neither do they consider it as a practice that is only good for a particular gender. Again, the planting of cover crops is a common practice in KZN to protect and augment the fertility of the soil. Cover crops such as oats, vetch, and triticale are planted as a winter cover in maize farms in KZN, while sorghum, although with lesser frequency, is planted as a summer cover crop [32].

The high level of acceptance for rotational cropping by farmers in the Mthonjaneni Municipality also confirms the assertion of Strachan [32] that crops such as cowpea, dry beans, and soybeans are often used in rotation with other crops and are intercropped with maize by farmers in KZN. The responses of the farmers further reveal that the efficient management of manure conformed very well with their cultural values, and therefore, they showed a high level of acceptance for it. The farmers in uMhlathuze Municipality, in addition to the planting of cover crops, highly accepted agroforestry and diet improvement for animals, implying that the practices are not in conflict with their cultural values, neither do they consider them as practices that are only good for a particular gender. This finding reveals a high acceptance for some CSA practices based on the perception of the farmers that they do not conflict with their cultural or gender norm supports the claim of Murray et al. [33]. Murray et al. [33] argued for the need to pay attention to gender issues in mainstreaming CSA technologies. Beyond the comparative analysis, the whole sample analysis revealed that farmers in KCDM highly accepted agroforestry, planting of cover crops and diet improvement for animals. These findings suggest that implementing the cultivation of cover crops, rotational cropping, agroforestry, in CSA programs, and projects for crop farmers and practices such as manure management and livestock diet improvement for livestock farmers will attract no cultural or social conflict in the study areas.

The use of wetlands showed a low level of acceptance among the farmers in both local municipalities, suggesting that some of the farmers considered the use of wetlands to conflict with their social or cultural values, or both. The farmers in Mthonjaneni Municipality also showed a low level of acceptance for soil conservation and diet improvement for animals, while their counterparts in the uMhlathuze Municipality, in addition to the use of wetlands, showed a low level of acceptance for conservation agriculture and planting of drought- and heat-tolerant crops. These findings suggest that careful consideration of the disposition and reaction of farmers to embracing the use of wetlands, soil conservation, and planting of drought- and heat-tolerant crops is essential for their inclusion for the implementation of CSA programs or projects.

Farmers' response to the technical compatibility of the identified CSA practices hangs on their perception of the ease of use of the practices. The survey farmers considered the use of organic manure,

rotational cropping, and mulching to be easy to practice and, as such, showed a high level of acceptance for them. This finding implies that the adaptation and implementation of these CSA practices will not be difficult for the farmers in the study areas. Besides, farmers in Mthonjaneni Municipality found it relatively easier to diversify their crop production, while those in uMhlathuze found planting of cover crops easy to practice. The combined analysis reveals that farmers in KCDM highly embraced the use of organic manure, rotational cropping, mulching, and planting of cover crops based on ease of adoption and practice in perspective.

　　Conservation agriculture, agroforestry, and soil conservation were poorly accepted, probably on the grounds of technical compatibility. The farmers' perceived them to be relatively challenging to adopt and practice. This situation could be because conservation agriculture, agroforestry, and soil conservation require some technical capacity and consistency that small-scale farmers may find it difficult to cope with. These findings imply that small-scale farmers need a lot of support and technical assistance for a successful mainstreaming of conservation agriculture, agroforestry, and soil conservation in the small-scale farming system. The findings on the acceptance or rejection of CSA practices based on technical compatibility agrees with the assertion that where the technical know-how on a specific CSA practice is limited, more information induces negative attitudes towards adaptation and implementation [34]. It is, therefore, needful to ensure farmers have sufficient information and skills on existing CSA technology. The need for sufficient information and skills corroborates the findings of Ojoko et al. [28] and Onyeneke et al. [29] that access to extension and advisory services play significant roles in CSA adaptation and implementation.

　　The use of organic manure, mulching and crop rotation were considered relatively cheap for adaptation and implementation and as a result, were highly accepted based on the financial implication. This finding reflects the assertion of Senyolo et al. [35], in their study on how the characteristics of agricultural techniques impact their acceptance, that affordability is a significant factor that can impact the possibilities of agricultural techniques being beneficial to farmers. Aryal et al. [30], while studying the factors affecting the adoption of CSA by farmers in the Indo-Gangetic plains of India, point out the significance of economic capital in successful CSA adoption and implementation. Onyeneke et al. [29] highlight the importance of farmers' income and access to credit in CSA adaptation. The significance of capital in CSA adaptation and implementation suggests that CSA practices that are not expensive to adopt and practice by farmers will be adequate for small-scale farmers with regards to effectiveness and efficiency and, therefore, will be suitable for promoting climate-smartness in the small-scale agricultural system.

　　Small-scale farmers need a lot of assurance and financial assistance to readily adopt CSA practices that pose a financial or economic threat to their agricultural production. This finding agrees with the opinion of Mwongera [3] that farmers will have more interest in the immediate benefits (to maximize yield and profit) they could enjoy from CSA than any long-term technical benefits CSA could offer. This finding also agrees with the assertion of Kahtri-Chhetri et al., [9] that farmers' preferences and willingness to pay for CSA technologies are significantly influenced by the costs of the technologies as communicated to them. Khectri-Chhetri et al. [9] argued that farmers might not be willing to invest in many technologies despite their potential benefits if they are perceived to be costly in terms of implementation. Furthermore, Long et al. [36] noted that the high cost of technology adaptation and implementation could dis-incentivise farmers in adapting agricultural techniques. As a result, adaptation policies should include the provision of financial assistance to enable farmers adapt various CSA technologies that are relevant to their conditions.

　　Agroforestry, rotational cropping and the use of organic manure were highly accepted with the reason that they are very environmentally friendly. The use of wetlands, planting of drought- and heat-tolerant crops, crop diversification, efficient manure management and diet improvement for animals were given low acceptance based on environmental friendliness. The pattern of acceptance shown by the farmers to the identified CSA practices reveals the understanding of the farmers on the potential of the identified CSA practices in contributing to the environment. The finding

of this study reflects the assertion of Mignouna et al. [8] that the perception of farmers on how compatible a technique is to their environment will likely influence their attitude towards the technique. Hence, there is the need to put into consideration the preconception of farmers on CSA practices and re-orient them if need be before mainstreaming CSA practices for adoption, particularly among small-scale farmers. Furthermore, farmers should be involved in the evaluation of CSA technology to find its suitability to their conditions and environment. The finding of this study revealing the significance of farmers' perception in CSA adaptation and implementation corroborates the submission of Abegunde et al. [21], Onyeneke et al. [29] and Vera et al. [37] on the need for farmers' access to information and extension services.

The pattern of acceptance across the social, technical, economic and environmental domains suggests that the use of organic manure could be easily implemented among the sampled farmers in Mthonjaneni Municipality, while the planting of cover crops could be easily implemented in uMhlathuze Municipality. The combined analysis suggests that the use of organic manure could be easily implemented in KCDM. This finding suggests that the use of organic manure could be given a good level of consideration for implementing CSA among small-scale farmers in KCDM. In contrast, based on the pattern of acceptance across the domains, it may be relatively difficult implementing diet improvement for animals for adaptation among the small-scale farmers in Mthonjaneni Municipality, while planting of drought- and heat-tolerant crops may be relatively difficult to implement in uMhlathuze and KCDM. This finding suggests that there is the need to raise more awareness on diet improvement for animals and planting of drought- and heat-tolerant crops among the small-scale farmers in KCDM. Furthermore, extension services should be driven towards enhancing the skills of the farmers in adapting diet improvement for their animals and planting drought- and heat-tolerant crops.

## 5. Conclusions

This paper, therefore, deviates from that conventional approach by focusing on the features of the CSA practices. The findings reveal how the farmers' perception of the attributes of the identified CSA practices can affect their disposition towards the acceptance of the practices. Analysis of the responses gathered from the farmers categorized the acceptance levels of the identified CSA practices into three; high, medium, and low categories of CSA acceptance. These categories were generated from a composite score built from the acceptance level index of the farmers. The acceptance level index is a reflection of the level of acceptance of each of the identified CSA practices among the respondents concerning their perspective on the social, technical, economic, and environmental compatibility of the practices.

This paper, based on its findings, argues that farmers' preferences for CSA practices and willingness to adopt are significantly different from one another based on the potential benefits and cost of the technologies as revealed to them. This paper further argues that farmers may not have enough will to implement many CSA technologies, even with their potential benefits. Given this, policies and programs aimed at mainstreaming CSA technologies and practices should pay adequate attention to site-specific factors, which have enough relevance to local conditions and farmers. Furthermore, policy designs and implementation should emphasize the important role of the provision of information on the available CSA technologies and practices. Also, financial plans and resources should be integrated into the CSA policy framework to assist farmers in the adoption of various CSA technologies that are suitable for their location. Finally, this study recommends that researchers conduct further empirical studies on the characteristics of CSA practices and how they influence CSA adaptation. Similar studies can be extended to other areas of South Africa to enhance the reliability and efficacy of the findings. Further studies can also focus on the role of institutional mechanisms and resource tenure in CSA adaptation and implementation.

**Author Contributions:** V.O.A. formulated the research investigation under the supervision of M.S. and A.O., V.O.A undertook raw data collection, data analysis, and draft manuscript compilation. Research scrutiny and scientific validation were carried out by M.S. and A.O. The outcome of the manuscript is the effort of all the authors. All authors have read and agreed to the published version of the manuscript.

**Funding:** This research was made possible with the financial assistance received from the "National Research Foundation of South Africa and The World Academy of Science (NRF – TWAS)" for NRF – TWAS African Renaissance Doctoral Scholarship (Grant UID 105460) and the "University of Zululand Research and Innovation Office."

**Acknowledgments:** The authors would like to acknowledge the National Research Fund (NRF), The World Academy of Science (TWAS), and the University of Zululand Research and Innovation Office for providing financial assistance for this research project. The authors also extend gratitude to municipal authorities of the Mthonjaneni and uMhlathuze local municipalities and King Cetshwayo District. Special thanks also go to the Department of Agriculture and Rural Development, as well as the farmers in Mthonjaneni and uMhlathuze municipalities for their facilitation of and participation in the study.

**Conflicts of Interest:** The authors declare no conflicts of interest.

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
