# Peer review of "Mainstreaming Climate-Smart Agriculture in Small-Scale Farming Systems: A Holistic Nonparametric Applicability Assessment in South Africa"

_agriculture, doi:10.3390/agriculture10030052_

Round 1

Reviewer 1 Report

General comments:

This article provides a useful investigation of a range of important factors influencing adoption of agricultural innovation in a South African context. It is generally well written and presented. Going beyond an understanding of adoption of innovation in terms of producer characteristics is valuable, although not novel. There seemed to be limited awareness of other literature already available on the many factors (beyond farmer characteristics) that influence adoption of innovations (e.g. Pannell et al 2006). While mention is made of "past studies", references are rarely provided. In general, there is limited use and referencing of past studies throughout the article.

While the results as presented are interesting, it would have been interesting to have had an analysis across domains i.e. revealing those practices which rated well across all or most of the social, economic, technical and environmental domains – so which could be relatively easily implemented. 

Specific comments:

Line 74: “Despite the urgency of understanding the applicability of CSA at farm-level, many CSA programs are deficient of information needed for successful CSA adoption among farmers”.

Requires reference

Line 79 “Given those as mentioned earlier…” Unclear: what are “those” you are referring to? Please clarify.

Line 81 First mention of KCDM in the article, please provide full name.

Line 86 Second mention of KCDM in the article, don’t need full name here

Line 86 First mention of KZN in the article, please provide full name

2.2. Research design: Explain what a “cross-sectional research design” is.

2.2.1 Conceptual framework

Refers to “past studies”, please provide examples

The final sentence of this paragraph, beginning “This line of thinking…” would also benefit from a reference

Line 117 “There are key elements involved in the transfer or adoption of technologies”

Please provide examples of these key elements (with reference to past studies that have identified them).

Line 135: “As earlier stated, Mthonjaneni and uMhlathuze Local Municipalities were selected, from the district, because of their agricultural potential”

However, you previously stated (Line 94) that “The district municipality (one out of ten) and local municipalities (two out of five) were then selected randomly.” So which was it, random selection or purposive selection based on agricultural potential?

2.4 Data Collection

Require dates of data collection – at least the month(s) and year

How was the survey administered: face-to-face with participants, in their homes or elsewhere?

Author Response

.

Reviewer 2 Report

This is a fine paper on an important topic, not only for African agriculture, but for many areas which will have to deal with climate change and the transition to a climate smart agriculture. The focus on farmers, particularly small holding farmers, is extremely relevant, as they are often absent in the discussion.

However, this paper is missing a better presentation of results, which is a significant defect in the paper.  Visual/graphical representations of results are important in such comparative studies so that readers may easily capture the significance of the results, without wading through a sea of text and charts of numbers on each case. 

Fortunately, it is relatively easy to remedy this and I would look forward to a next version with appropriate graphics/visuals. 

Also, please review again the text for English usage. Please find a few suggestions: 

l.46 delete extra "and" before "socio-"

l.47 "influences" not "influencers"?

l.55 "is" not "gets"

l.154 "account" not cognizance

l.159-161 repetitive text on the testing of questions. Please revise.

l.288 gender "sensitivity"? Perhaps the authors would like to introduce the issue of gender upfront in the article?

Author Response

.

Reviewer 3 Report

Dear authors,

please find my comments within the annotated PDF attached to this report.

Kind regards.

Author Response

.

Round 2

Reviewer 3 Report

The authors successfully revised the manuscript.